**Data Availability Statement:** All relevant data are within the manuscript and its Supporting information files.

# Identifying wastewater management tradeoffs: Costs, nearshore water quality, and implications for marine coastal ecosystems in Kona, Hawai'i

**Christopher A. Wada[1,2], Kimberly M. Burnett[1,2]\*, Brytne K. Okuhata[3], Jade M. S. Delevaux[4], Henrietta Dulai[2,3], Aly I. El-Kadi[2,3], Veronica Gibson[5], Celia Smith[5], Leah L. Bremer[1,2]**

1 University of Hawai'i Economic Research Organization, Honolulu, Hawai'i, United States of America,
2 Water Resources Research Center, University of Hawai'i at Mānoa, Honolulu, Hawai'i, United States of America, 3 Department of Earth Sciences, University of Hawai'i at Mānoa, Honolulu, Hawai'i, United States of America, 4 The Natural Capital Project, Stanford University, Stanford, California, United States of America, 5 Department of Botany, University of Hawai'i at Mānoa, Honolulu, Hawai'i, United States of America

\* kburnett@hawaii.edu

## Abstract

Untreated and minimally treated wastewater discharged into the environment have the potential to adversely affect groundwater dependent ecosystems and nearshore marine health. Addressing this issue requires a systems approach that links land use and wastewater management decisions to potential impacts on the nearshore marine environment via changes in water quality and quantity. To that end, a framework was developed to assess decisions that have cascading effects across multiple elements of the ridge-to-reef system. In an application to Kona (Hawai'i, USA), eight land use and wastewater management scenarios were evaluated in terms of wastewater system upgrade costs and wastewater related nutrient loads in groundwater, which eventually discharge to nearshore waters, resulting in potential impacts to marine habitat quality. Without any upgrades of cesspools or the existing wastewater treatment plant (WWTP), discharges of nutrients are expected to increase substantially with permitted development, with potential detrimental impacts to the marine environment. Results suggest that converting all of the existing cesspools to aerobic treatment units (ATU) and upgrading the existing WWTP to R-1 quality provide the highest protection to nearshore marine habitat at a cost of $569 million in present value terms. Other wastewater management options were less effective but also less costly. For example, targeted cesspool conversion (a combination of septic and ATU installation) in conjunction with the WWTP upgrade still provided a substantial reduction in nutrients and potential impacts to marine habitat quality relative to the present situation at a price point roughly $100 million less than the entirely ATU option. Of note, results were more sensitive to the inclusion of the WWTP upgrade option than they were to assumptions regarding the efficiency of the cesspool conversion technologies. The model outputs also suggest that the spatial distribution of potential impacts should be carefully considered when comparing different wastewater management scenarios. When evaluated separately, the WWTP option reduced total nutrients

**Funding:** Funding for this project came from the National Science Foundation's Research Infrastructure Improvement Award (RII) Track-1: 'Ike Wai: Securing Hawai'i's Water Future Award #OIA-1557349 and USGS Water Resources Research Institute Program grant number G16AP00049 BY5 "Linking watershed and groundwater management to groundwater dependent ecosystems and their linked ecological, cultural, and socio-economic values." The funders had no role in study design, data collection and analysis, decision to publish, or preparation of the manuscript.

**Competing interests:** The authors have declared that no competing interests exist.

by more than the targeted cesspool conversion option at a fraction of the cost. However, potential improvements in marine habitat quality only occurred in the immediate vicinity of the WWTP, whereas the benefits under targeted cesspool conversion were more evenly distributed along the coast.

## 1. Introduction

Sufficient and targeted wastewater management is a key component in supporting and protecting groundwater dependent ecosystems such as wetlands and spring-fed agricultural systems, and nearshore marine environments around the world [1–3]. Wastewater is a significant threat to groundwater dependent and other coastal ecosystems, including coral reefs [4–6], yet the impacts of wastewater on valued marine ecosystems is rarely accounted for in decision making [7]. An increasing number of jurisdictions have introduced legislation to mandate upgrades of older cesspool technologies to newer, more advanced systems in order to reduce the delivery of nutrients to nearby groundwater aquifers and connected groundwater dependent ecosystems [6]. Implementing and operationalizing such legislation requires understanding both the benefits of various wastewater management strategies and technologies as well as the differing costs associated with these options.

Ecologically, unpolluted submarine groundwater discharge (SGD) is a natural component of many nearshore coral reef ecosystems, and supports native and endemic nearshore species which rely on SGD-influenced habitat for survival. Healthy coral reefs provide a suite of societal benefits to important income and livelihood sectors including by supporting fisheries and food security, and by enhancing recreation and tourism value [8–10]. Coral reefs also provide important coastal protection services, which enhances resilience to climate change [11, 12]. SGD delivers cool, fresh, nutrient rich groundwater to nearshore reefs, which drives productivity when taken up by macro and micro algal species, and creates areas of estuarine habitat on which marine plant, animal, and invertebrate species rely in various capacities [8, 13]. Declines in SGD have been shown to have negative effects on nearshore biota [4]. Worldwide, anthropogenic nutrient loading of SGD has been linked to macroalgal blooms, shifts from coral to macroalgal dominated ecosystems, harmful phytoplankton blooms and eutrophication [5]. Such threats to coral reefs can have devastating impacts on coral reef ecology and the array of ecosystem services provided by these systems [8, 14].

Hawai'i has become an important focal point for wastewater management within the United States due to increasing concern about threats to coral reef ecosystems of high value for tourism, recreation, fisheries, and cultural connection to place [15, 16]. Moreover, the connection between existing wastewater technologies and nutrient loading to the nearshore has also been extensively studied in Hawai'i [17–23]. There are approximately 88,000 cesspools across the state, releasing over 200,000 $m^3$ of wastewater per day to the environment [24]. Much of that wastewater percolates into underlying groundwater systems and eventually enters nearshore marine ecosystems via SGD. In response to concerns that cesspools pose a potential environmental and public health risk, the Hawai'i State Legislature passed Act 125 in 2017, which states that all cesspools in the State, unless granted exemption, must be upgraded to a septic or aerobic treatment unit (ATU), or connected to a sewer system by January 1, 2050. A Cesspool Conversion Working Group (CCWG) was established in the following year under Act 132 to assist in facilitation of Act 125 [6]. The management scenarios presented in the

current paper were developed through an iterative discussion process with the CCWG with the goal of generating research results that provide utility to decision makers.

The State Department of Health identified 6,500 cesspools in the coastal Kailua-Kona area of Hawai'i Island as having the potential to negatively impact sensitive coastal waters [24, 25]. While not expected to pose a significant risk to drinking water sources, these cesspools were identified as having negative impacts on nearshore marine water quality and linked groundwater-dependent ecosystems (GDE). GDEs have historically served as important sources of water and food to Kona's coastal communities and continue to be highly valued today for their cultural uses and provision of a wide variety of ecosystem services [26, 27]. The importance of GDEs and the SGD that supports them is not unique to Hawai'i, however, as a number of studies have documented the many societal values of these systems worldwide [28]. Thus, developing a framework to evaluate the potential impacts of land use and wastewater management decisions on nearshore ecosystems will not only help to inform local policy, but will also have broader applications for management and ecosystem implications.

To that end, this paper discusses a systems approach that was developed to link land use decisions (development, water use, wastewater treatment), nutrient fate and transport processes in groundwater, and nearshore marine health. Whereas a single model built for any of these individual components would likely be sufficient to test various hypotheses about that particular component, our developed ridge-to-reef approach provides a means for assessing decisions that have cascading effects across multiple elements of the system. Several terrestrial management scenarios are evaluated in terms of the nutrients nitrogen (N) and phosphorus (P) discharged to nearshore waters, potential impacts to marine habitat quality, and costs. Results are then presented in three ways: (1) maps that show the spatial distribution of impacts for each scenario, (2) spider diagrams that illustrate tradeoffs between multiple management objectives based on aggregated spatial outputs, and (3) a cost-benefit analysis, where benefits are measured in physical units and costs in dollars.

## 2. Materials and methods

### 2.1 Study site description

The Keauhou aquifer (426 km$^2$) encompasses the southern flank of Hualālai volcano, located in the northern Kona district of Hawai'i Island (Fig 1). Like most volcanic island aquifers, groundwater is the major source of fresh drinking water for residents of the Kona community [29]. The Keauhou aquifer has an anomalous hydraulic head gradient approximately six kilometers inland, where water levels drastically increase from 1 m to >100 m above mean sea level. A geologic subsurface structure, referred to as the high-low divide, is believed to prevent high-level groundwater from quickly flowing to the coast, therefore creating a productive high-level aquifer that supplies approximately 25% of Kona's major pumping wells [30]. The basal aquifer is considered to be the area where water levels are <5 m above mean sea level (150 km$^2$), extending from the high-low divide to the western coastline.

### 2.2 Modeling approach overview

The ridge-to-reef framework consists of three models coupled with an assessment of potential impacts to the marine habitat (see Methods sections below for greater detail). Fig 2 provides a high-level graphical overview of how the different elements of the ridge-to-reef framework are connected. Groundwater recharge (Fig 2A) [31] feeds into the groundwater model (Fig 2B), while land use and wastewater management scenarios (Fig 2D) are concurrently applied to the groundwater model domain to generate groundwater nutrient concentrations (Fig 2C) [32]. Using the simulated nutrient concentrations [32] coupled with spatial information about

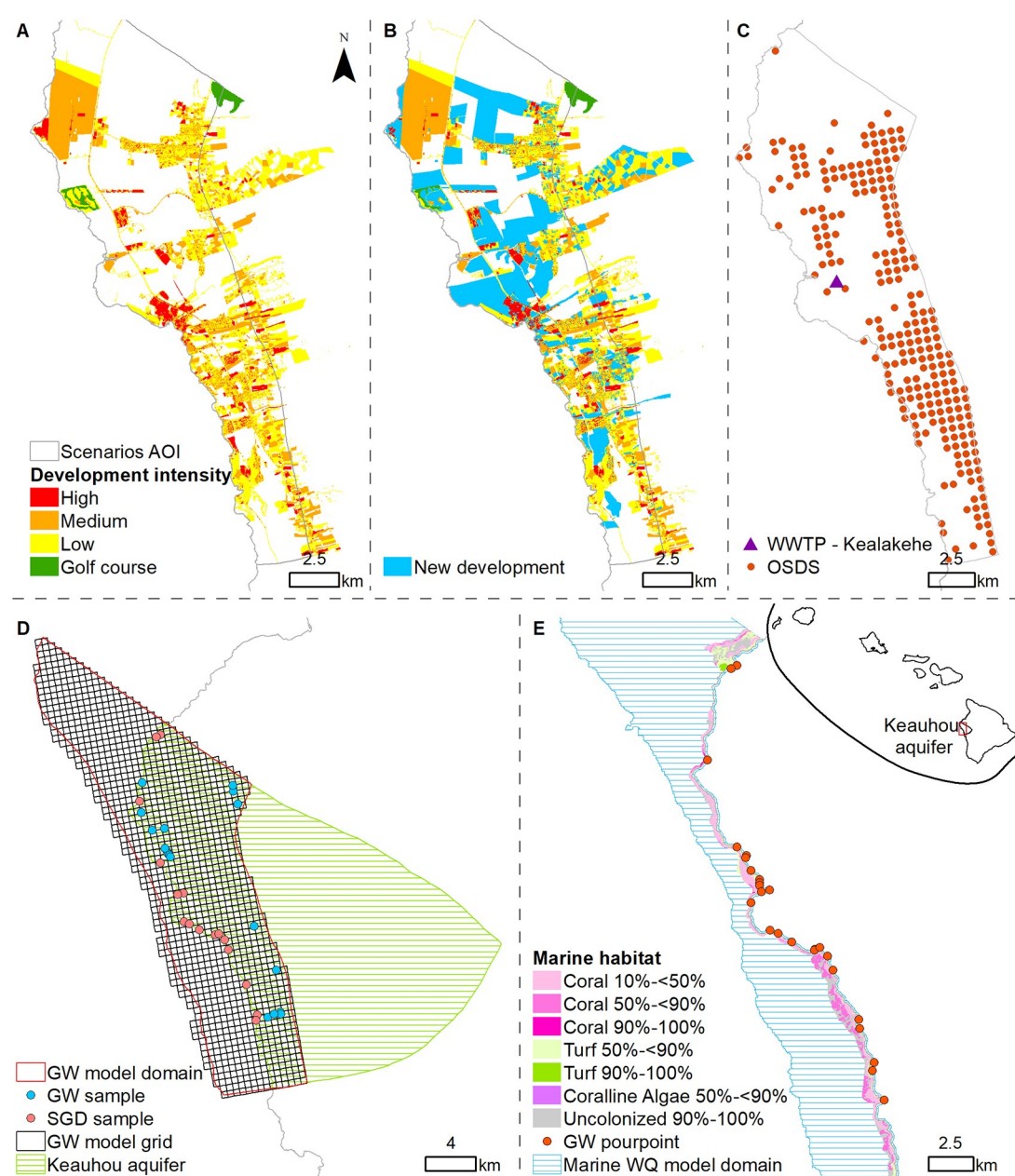

**Fig 1. Study site.** (A) Current land cover; (B) Future permitted land cover; (C) Location of the WWTP and model grid cells containing one or more cesspools; (D) Groundwater model domain and empirical data; (E) Marine water quality model domain, pourpoints, and marine habitat.

depth [33] and wave power [34] (Fig 2F), the marine water quality model (Fig 2E) generates maps of changes in marine water quality in terms of nutrient loads (kg/yr) (Fig 2H) and feeds into the assessment of potential impacts to the marine habitat [35] (Fig 2G). Finally, this approach produces maps of potential marine habitat impacts, and all of the model outputs (i.e., change in GW and marine WQ) (Fig 2H) are compared to one another using the costs associated with the considered management scenarios.

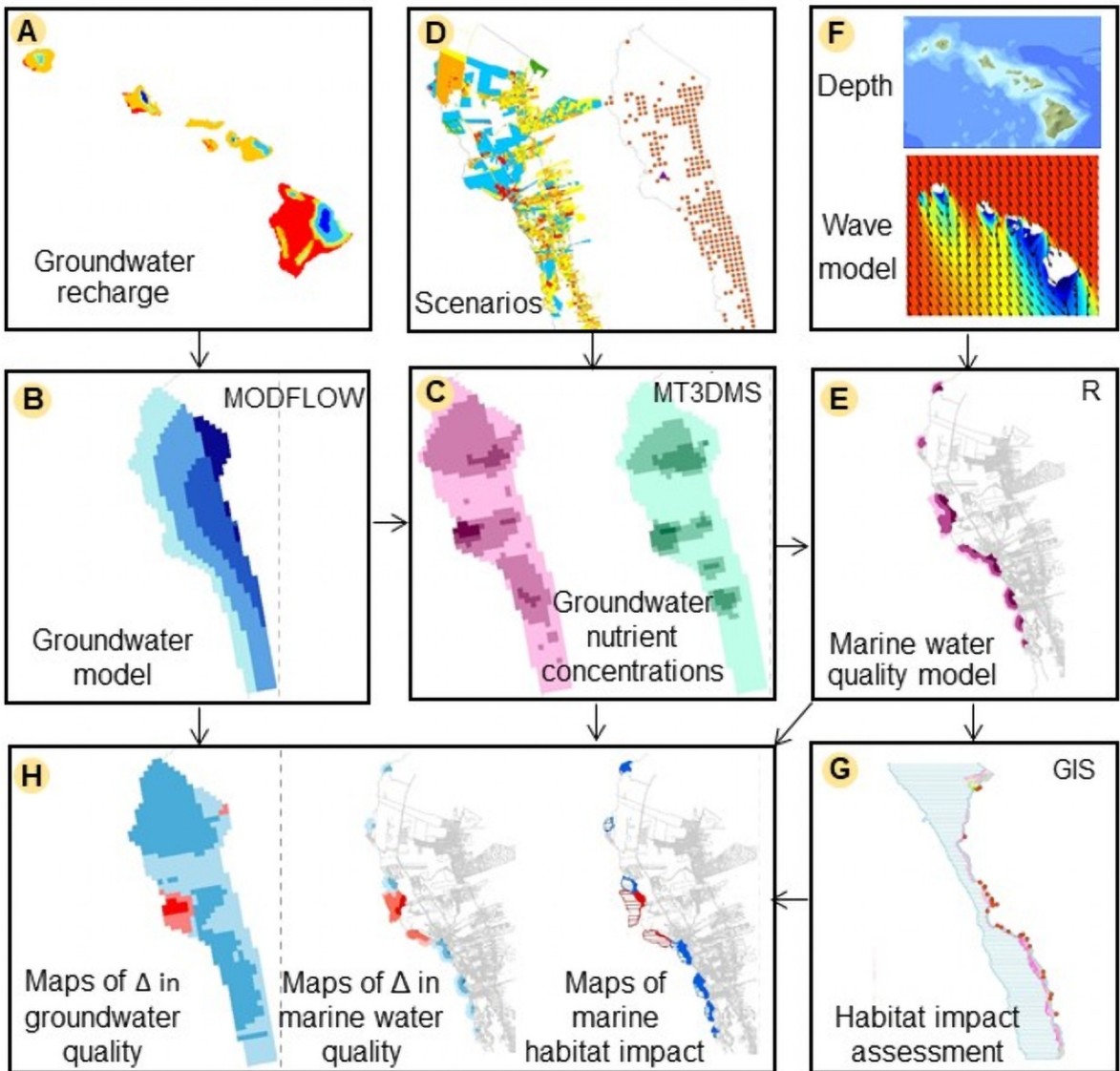

**Fig 2.** Connections between modeling elements: (A) Groundwater recharge is an input to (B) the groundwater budget model, and (C) the groundwater nutrient concentration model coupled with (D) land use, wastewater management scenarios, and water demand. The nutrient concentrations serve as inputs to (E) the marine water quality model and (G) marine habitat potential impact assessment. (H) The outputs of the analysis consist of maps of change in groundwater quality, marine water quality and potential impact to the marine habitat. (The wave model image in panel F is reprinted from [36] under a CC BY license, with permission from Charles Fletcher, original copyright 2009).

## 2.3 Wastewater management and land cover change scenarios

A total of eight scenarios were developed by combining different assumptions about future land development, water demand, cesspool conversion extent and technologies, and municipal wastewater treatment plant (WWTP) upgrade status. The location of the WWTP and current aggregated cesspools are shown in Fig 1C. The first scenario describes present-day conditions, with none of the cesspools converted and the WWTP not upgraded. The future permitted build out scenario simulates the case where land development occurs but no wastewater mitigation strategies are undertaken and all new development is served by the existing WWTP. The remaining six scenarios describe different wastewater management options under the

**Table 1. Description of wastewater management scenarios.**

| No. | Scenario name | Land use | Cesspool conversion* | WWTP upgrade |
|---|---|---|---|---|
| 1 | Present | Current land use | None | No |
| 2 | Future permitted (inaction) | Addition of permitted development | None | No |
| 3 | All ATU upgrade (+ WWTP upgrade) | Current land use | All ATU (high efficiency) | Yes (90% mass load reduction) |
| 4 | WWTP upgrade | Addition of permited development | None | Yes (90% mass load reduction) |
| 5 | Target low efficiency | Addition of permitted development | Targeted septic & ATU (low efficiency) | No |
| 6 | Target high efficiency | Addition of permitted development | Targeted septic & ATU (high efficiency) | No |
| 7 | Target low efficiency + WWTP upgrade | Addition of permitted development | Targeted septic & ATU (low efficiency) | Yes (90% mass load reduction) |
| 8 | Target high efficiency + WWTP upgrade | Addition of permitted development | Targeted septic & ATU (high efficiency) | Yes (90% mass load reduction) |

*Cesspool conversion efficiencies are described in Table 2.

assumption that the permitted build out occurs in the future. For the options with targeted conversion (Scenarios 5–8 in Table 1), risk scores determine whether cesspools in a particular area are converted to either septic or ATU systems. Risk scores were calculated with a weight and rating method using geographical mapping and numerical modeling [25]. Various factors, such as proximity to the shoreline, distance to groundwater, slope, and soil type were weighted based on their relative contribution to the OSDS risk. The efficiency ratings (low or high) denote how effective the upgrade technology is assumed to be at removing nutrients. Assumptions underlying each scenario are described in greater detail in the remainder of this section.

**2.3.1. Land and water use.** Current land cover is shown in Fig 1A. Current water consumption is based on pumping rates reported to the Commission on Water Resource Management (CWRM). On average from 1990–2017, approximately 42,000 $m^3$/d of water was pumped from the Keauhou basal aquifer, while approximately 16,310 $m^3$/d was pumped from high-level wells located upgradient of the basal aquifer and outside of the model domain. Future development (Fig 1B) is based on allocated water permits, as indicated in the Kona Water Use Development Plan (WUDP), where each new residential unit was designated a water demand of 1.5 $m^3$/d [37]. In all of the future scenarios, future water demand is assumed to be supplied entirely by pumping from high-level wells. This results in a 64,080 $m^3$/d reduction in influx from the upland boundary condition of the groundwater model. We further assume that 50% of the future water demand will be used for outdoor purposes [38], while the remaining water (32,040 $m^3$/d) is used indoors before being sent to the WWTP. Nutrient concentrations were applied to water flowing through each land parcel based on development type and intensity. Based on groundwater measurements from Keauhou pumping wells, nutrients released from natural land were assigned N and P concentrations of 1.0 mg/L and 0.1 mg/L, respectively. Groundwater recharge affected by golf courses (locations obtained from the Hawai'i Statewide GIS Program: http://planning.hawaii.gov/gis/download-gis-data/) were assigned N and P concentrations of 7.59 mg/L and 0.54 mg/L, respectively [39]. Urban development was further divided into three intensity levels (low, medium, high) according to the 2005 Hawai'i Coastal Change Analysis Program [40]. The low, medium, and high urban development intensities were assigned N concentrations in groundwater recharge of 1.13, 1.07, and 1.02 mg/L, respectively, and P concentrations of 0.107, 0.104, and 0.101 mg/L, respectively [41]. Where new development is anticipated in areas of previously-assigned natural land, nutrient concentrations associated with medium intensity development were assigned to recharging groundwaters. New development anticipated in previously developed areas were assigned new nutrient concentrations for groundwater recharge based on anticipated water demand. If anticipated water demand was 1.5 $m^3$/d (i.e., one residential unit), nutrient

concentrations were assumed to remain the same, but if water demand was >1.5 m³/d, nutrient concentrations were assumed to increase from medium to high intensity.

**2.3.2. Wastewater treatment.** A number of cesspool conversion technologies, each with its own set of costs and range of nutrient removal efficacy, could be implemented to meet the requirements of Act 125. In general, most upgrade options consist of a treatment system and a disposal system. The former provides primary or secondary treatment of household wastewater. Examples include septic tanks and ATU, some of which are capable of denitrification. The latter are paired with a treatment system to dispose of the treated wastewater. Examples include absorption systems and seepage pits. Based on discussions with the Cesspool Conversion Working Group, the model assigns one of two technologies to existing cesspools depending on risk scores that incorporate groundwater and drinking water risk, stream and watershed risk, coastal water risk, and risk related to soil type [25]. Cesspools located within the same grid cells are aggregated into a single representative cesspool for each cell with an averaged risk score (Fig 1C). OSDS risk scores range from 7 to 18 [25], and the median score of 12 was selected as the conversion threshold score (see S1 Fig in the Supporting Information section). Aggregated cesspools with an average risk score <12 are upgraded to a septic tank and absorption bed/trench system in the targeted conversion scenarios. Aggregated cesspools with an average risk score ≥12 are upgraded to an ATU paired with an absorption system in the targeted and all-ATU scenarios. The majority of OSDS units (52%) are assigned a risk score of 12. Therefore, with a conversion threshold score of 12, 60% of the OSDS units will be upgraded to an ATU while 40% will be upgraded to a septic tank. If a risk score of 11 was instead selected as the conversion threshold, 84% of the OSDS units would be upgraded to an ATU system, whereas only 8% would be converted with a threshold risk score of 13. It is therefore important to acknowledge that total nutrient mass load can easily fluctuate, depending on the proportion of OSDS units that are converted to ATU systems. Given that the effectiveness of a particular conversion technology will depend on a variety of site-specific factors, low and high efficiency average nutrient removal rates are considered for both septic and ATU upgrades, in accordance with the range of values presented in [42]. Nutrient concentration reduction assumptions for cesspool upgrade options are summarized in Table 2.

The Kealakehe WWTP, the only municipal WWTP in the study area, currently treats approximately 6,400 m³ of sewage per day from the North Kona sewerage system to secondary treatment classification through the use of aerated lagoons. The treated effluent is disposed of in a nearby percolation basin (Fig 1C), after which it eventually discharges into coastal waters. Typical nutrient concentrations of this effluent are 22.6 mg/L for total N and 7.01 mg/L for total P [38]. Given concerns about the potential impacts the released effluent may have on nearshore water quality, an upgrade to the WWTP has been proposed in recent years. Through a combination of R-1 treatment, subsurface flow constructed wetlands, and soil aquifer

**Table 2. Assumptions regarding nutrient concentrations in released effluent for wastewater treatment alternatives.**

| | N (mg/L) | P (mg/L) | Source |
|---|---|---|---|
| Cesspools | 60.5 | 16.5 | [42] |
| Septic upgrade (low efficiency) | 45.4 (25% reduction) | 16.5 (0% reduction) | [41] |
| Septic upgrade (high efficiency) | 27.2 (55% reduction) | 13.2 (20% reduction) | [41] |
| ATU upgrade (low efficiency) | 27.2 (55% reduction) | 14.9 (10% reduction) | [41] |
| ATU upgrade (high efficiency) | 9.1 (85% reduction) | 11.6 (30% reduction) | [41] |
| Current WWTP | 22.6 | 7.0 | [43] |
| Upgraded WWTP | 2.3 (90% reduction) | 0.70 (90% reduction) | [43] |

treatment (SAT), the upgrade is projected to reduce N and P by up to 90% [43]. Assumed nutrient concentrations in released effluent for the current and proposed WWTP upgrade are presented in Table 2. For management scenarios with increased future urban development, we assume that all new residential units are connected to the WWTP and that the amount of sewage entering the WWTP is consequently increased by 50% of projected future water demand, while the remaining 50% is allocated to outdoor water uses [37].

## 2.4 Groundwater model

This study utilized a three-dimensional, density-dependent, multi-species numerical groundwater model that was previously developed [32] with the simulation program SEAWAT [44, 45] within the GMS software interface (https://www.aquaveo.com/software/gms-models-utilities). The numerical model spans over the Keauhou basal aquifer and offshore, and consists of 21 layers (26,323 cells total) with a top elevation that follows local topography and bathymetry and a flat bottom elevation of 550 m below mean sea level (see S2 Fig the Supporting Information section). The bottom elevation of the first layer is located 1 m below mean sea level to ensure dry cells were not produced and the remaining layers increase in thickness, where the uppermost layers are thinnest. The horizontal spacing of the grid cells are a consistent 490 m by 490 m (Fig 1B). Recharge was estimated based on the results from the USGS Hawaiʻi Island water budget model [31]. The top two layers of the eastern boundary condition were assigned a constant flow rate of 388,399 $m^3$/d to account for the upland recharge and pumping, and specified salinity, N, and P concentrations of 0.26, 0.001 g/L, and 0.0001 g/L, respectively, as measured in groundwater wells, to account for upland inflow conditions. The land-use nutrient concentrations previously described were intersected with the recharge spatial distribution and applied as recharge concentrations. Aggregated OSDS and the Kealakehe WWTP were applied as point-source mass loads.

Efforts to simulate SGD as a diffusive ocean-interface process as a default approach in the model were not successful and produced results that were drastically different from observed discharge patterns. SGD mainly occurs as coastal springs with no evidence of significant diffusive fluxes [45]. To improve on results, and following similar approaches taken in previous studies [30, 46], discrete SGD springs identified by [45] were simulated as point drains (head-dependent sinks). Here, it is assumed that the conductance of each spring is proportional to spring discharge. Therefore, each spring has an individual conductance related to the measured discharge, which varies across the complex aquifer coastline. Nineteen of the 27 springs were assigned conductance values based on measured discharge rates, with a total simulated discharge of ~100,000 $m^3$/d. Discharge rates were not measured for the remaining eight springs, so average rates were applied, thus increasing the total simulated discharge to ~140,000 $m^3$/d. The model was calibrated with 54 well water-level measurements that were obtained from CWRM and were measured between 1944 and 2008. Salinity, N, and P concentrations were calibrated with 20 well measurements [47] and 13 SGD measurements (Fig 1B).

## 2.5 Marine water quality model

To link the groundwater nutrient loads to the potential impact on coral reef habitat, the nutrient flux (kg/yr) was diffused into the marine environment (60 x 60 m) from the coastal springs, from here on referred to as pourpoints for the purpose of the water quality modeling. The pourpoints represent the SGD for each management scenario. A previously developed marine water quality model tested in Hawaiʻi [39] was applied, which was adapted to account for diffusion from point sources (Fig 1E). First, a diffusion factor layer was created that represents the impedance of moving planimetrically through each cell from each groundwater pourpoint

using a composite of two marine drivers known to affect diffusion (depth [m] and wave power [KW/m]) [48, 49]. Then, the spread of nutrient loads into the marine environment from each pourpoint was modeled using a decay function (see Eq 1), which assigned a portion of the remaining nutrient loads from the previous cell to all adjacent cells based on the diffusion factor layer until a maximum distance of 1.2 km from the shoreline was reached [50, 51]. This threshold was based on measurement of infrared imagery from the study site in ArcGIS and consultation with local experts.

$$N_i = n \times e^\wedge(-c^{2/Dc}) \tag{1}$$

where $N_i$ = Grid cell value for diffused nutrient flux (kg/yr) per pourpoint $i$, n = Nutrient load (kg/yr) at each pourpoint (obtained from the GW model), c = diffusion factor layer value at each grid cell (unitless), and Dc = distance threshold from the shore for each pourpoint (set to 1.2 km).

Then, all the individual nutrient plumes from each pourpoint were summed to obtain the aggregated nutrient plume at the study site scale, per management scenario. Note that this diffusive approach to modeling submarine groundwater nutrient discharge accounts for wrapping around coastal features and captures the nearshore advection forces that push nutrients in specific directions [35]. For each scenario, the relative change to the current conditions were calculated to identify areas where water quality changes and may impact the marine habitat. The results displayed here use the geometrical intervals from the "best" (scenario 3, full upgrade) and "worst" (scenario 2; no upgrade) case scenarios to identify the breaks while also highlighting the differences across scenarios to enable comparison.

## 2.6 Coral reef habitat potential impact assessment

For the coral reef habitat potential impact assessment under the scenarios considered here, the NOAA benthic habitat map [35] was used to identify and quantify areas of coral reef habitat (ha) exposed to a change in coastal water quality (as modeled by the marine quality model), i.e., the response of coral reefs to change in nutrient exposure was not explicitly modeled. For the purpose of this analysis, only live coral cover (879.6 ha) and turf algae cover (332.5 ha) were considered because they are the two most abundant live cover habitat classes in the study area and previous research has shown that they are sensitive to changes in water quality and/or compete with one another when subject to climate change bleaching impacts [5, 39, 40]; the assessment ignored 'uncolonized', 'coralline algae', 'unclassified' and 'unknown' classes [38] (Fig 1E). Aside from abundance relative to other live cover habitat classes, there were two main reasons for the focus on coral and turf algae cover. First, previous research has shown that an increase in nutrient loads will likely result in an increase in turf algal cover and vice versa [52–54]. Second, research has also shown that turf algae respond quickly to environmental changes and are more competitive for space than coral under increased nutrient conditions. These attributes could favor turf algae and hinder corals from recovering after bleaching events [55, 56].

## 2.7 Economic costs

As discussed previously, each cesspool within the model domain is converted to either a septic tank + absorption system or an ATU + absorption system, depending on its risk score. The present value or life cycle cost of each system consists of the one-time installation cost, which varies by capacity (number of bedrooms served), and annual operation and maintenance (O&M) costs, which are assumed to be independent of system capacity. Following a previous study that assessed the costs of cesspool upgrade options in Maui (Hawai'i) [41], the septic

**Table 3. Life cycle costs for septic and ATU systems.**

| System | 1 BR | 2 BR | 3 BR | 4 BR | 5 BR |
|---|---|---|---|---|---|
| Septic | $27,522 | $28,722 | $29,322 | $30,922 | $32,422 |
| ATU | $51,787 | $52,801 | $55,844 | $58,887 | $65,407 |

option is assumed to have an annual O&M cost of $400 with a 60-year system replacement interval and the ATU option is assumed to have an annual O&M cost of $700 with a 30-year system replacement interval. The life cycle cost over 30 years of each system is presented in Table 3 for a range of capacities (1–5 bedroom (BR)), assuming a discount rate of 2.8%.

Because the life cycle costs of both systems vary substantially by installed capacity, the size of each potential system upgrade was assigned based on the characteristics of parcels that are currently identified as having one or more cesspools. Using the State's GIS layer for OSDS on the island of Hawai'i [57], we first assigned a system capacity equal to the number of bedrooms for all parcels in which the total number of bedrooms was less than or equal to five. Parcels with zero bedrooms were assigned a 1-BR capacity. For parcels with greater than five bedrooms, the total number of structures on the property was divided by the total number of bedrooms. Capacities were then assigned based on the average number of bedrooms per structure. For example, on a parcel with 100 structures and 200 bedrooms, each structure was assumed to have two bedrooms, meaning 100 2-BR systems would be installed. Using this approach, 233, 1,334, 4,088, 973, and 625 systems were assigned to 1-BR, 2-BR, 3-BR, 4-BR, and 5-BR capacities respectively, summing to a total of 7,253 systems.

For each scenario, all, none, or a subset of the existing cesspools are converted to septic or ATU systems based on their risk scores and estimated required capacities. S1 Table in the Supporting Information section summarizes the conversion totals for the eight scenarios. Total cesspool conversion costs were then calculated for each management scenario by multiplying the appropriate life cycle cost by the number of systems of each capacity requiring conversion for both septic and ATU upgrades.

The proposed Kealakehe WWTP R-1 upgrade is estimated to cost roughly $160 million, including $35 million for soil aquifer treatment, $25 million for a subsurface wetland, and $20 million for piping, among other expenses [58]. Although the decision to invest in the upgrade is still up for debate—the Hawai'i County's Department of Environmental Management has continued to research less costly alternatives like expanding infrastructure to deliver wastewater treated at the (current) R-2 level to golf courses and agricultural users for irrigation—the analysis presented in the current paper considers only the potential nearshore water quality impacts and costs of the proposed R-1 upgrade.

## 2.8 Analysis of tradeoffs

Impacts of different management strategies, measured in terms of simulated changes in nutrients and potential changes to marine habitat quality, are spatially heterogeneous because the model and represented system change are, by design, not spatially uniform. On one hand, this ensures that nutrient or potential marine habitat quality hotspots are not inadvertently "smoothed out" as might occur when using a spatially lumped approach that focuses on aggregated or averaged values. On the other hand, with higher spatial granularity, comparing impacts of multiple scenarios simultaneously can be challenging, as it requires evaluating distributions of impacts rather than single values. With that in mind, spider diagrams [59] summarizing results across each scenario were generated to supplement and aid in interpretation

of the model's detailed spatial results. To create the spider diagrams, each of four metrics (N reduction, P reduction, potential change in marine habitat quality, present value cost) were ranked ordinally for all scenarios except the current condition, as the current condition is the base against which the nutrient and marine habitat results are compared. For each metric, the scenario with the top ranked output was assigned a value of seven, the second ranked output was assigned a value of six, and so forth. For a given scenario, scores for each metric were then plotted on a set of concentric circles, where the innermost circle represents the minimum score (one), the outermost circle represents the maximum score (seven), and each cardinal direction represents one of the metrics being evaluated. The area of the polygon generated by connecting the plotted points for each scenario represents an aggregate score that can be directly compared.

A cost-benefit analysis was also undertaken, where benefits were measured in physical units and costs in dollars, to further inform prioritization of the management scenarios. Simulated N, P, and marine habitat quality for each management scenario were compared to the future permitted scenario, and the difference was divided by the respective scenario's cost in order to estimate the conservation benefit generated per million dollars spent on a particular management action [60]. That is, each potential future was compared to the inaction scenario, wherein future land development occurs but no action is taken to address the wastewater management problem, to determine the return on investment (ROI) of each candidate management action. As previously noted, caution is warranted when interpreting results generated using aggregated values given the spatial nature of the management problem. However, ROI estimates when viewed in conjunction with spatial impact maps and spider diagrams may allow for more concrete policy recommendations.

## 3. Results

### 3.1 Groundwater & marine water quality

Groundwater nutrient concentrations vary in severity across the Keauhou basal aquifer. Under present conditions, nutrient concentrations are highest surrounding the Kealakehe WWTP (Figs 3A and 4A). Nutrient concentrations are also relatively elevated north and south of the WWTP, in areas with high OSDS quantities. Under future permitted conditions, overall nutrient concentrations generally increase, but the most significant change is around the WWTP due to the drastic increase in nutrient mass load from the additional anticipated effluent discharge (Figs 3B and 4B). If the WWTP is upgraded, however, the nutrient concentrations surrounding the WWTP will decrease (Figs 3D and 4D). If OSDS are converted to high efficiency ATU and septic units, lower nutrient concentrations are seen across the aquifer in comparison to low efficiency conversions. This is particularly noticeable in areas located south of the WWTP (Figs 3E–3H and 4E–4H). This is likely because a higher proportion of OSDS south of the WWTP were assigned risk scores ≥12, therefore being converted to ATU rather than septic systems.

The higher number of SGD pourpoints representing prolific SGD springs to the south of Kailua Kona and the WWTP results in more nutrient discharge along the southern coast. Because of this spatial variation, upgrading the cesspools across the study area results in more nutrient reduction along the southern coastline of the site and less along the northern coastline (Figs 5E–5H and 6E–6H). The nutrient load discharge reduction is lower under the target low efficiency (Figs 5E and 6E) compared to the target high efficiency scenario (Figs 5F and 6F). Under both target low and high efficiency scenarios, the models show an increase in nutrient load export downstream from the WWTP due to the increase in sewer connections assumed for the permitted build out (Figs 5E–5F and 6E-6F). When the WWTP is upgraded and

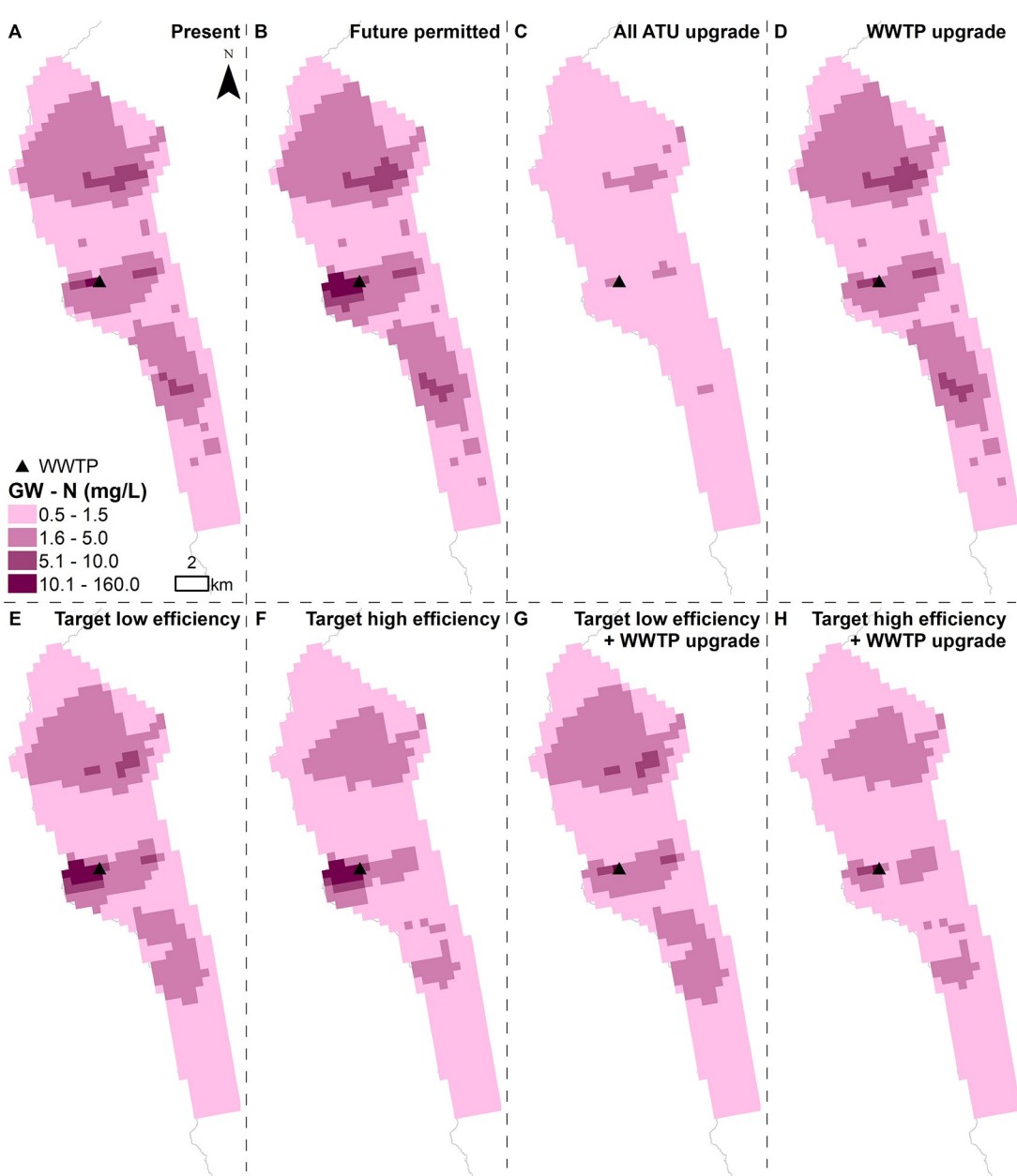

**Fig 3. Simulated groundwater N concentrations for (A) current conditions and (B-H) all scenarios.** Maps display results from the first layer of the groundwater model, where the bottom of the layer is set to 1 m below mean sea level.

coupled with target low or high efficiency OSDS, the models show a nutrient load reduction across the entire modeled area (Figs 5G–5H and 6G–6H).

## 3.2 Coral reef ecosystem potential impacts

The future permitted buildout results in the largest increase in nutrient load discharge into the ocean (Fig 7B). The all-ATU upgrade scenario results in the largest decrease in nutrient loads discharging into the ocean and therefore the largest area of marine habitat being less exposed to land-based nutrients (Fig 7C). Due to the spatial variation in the SGD along the coast, the

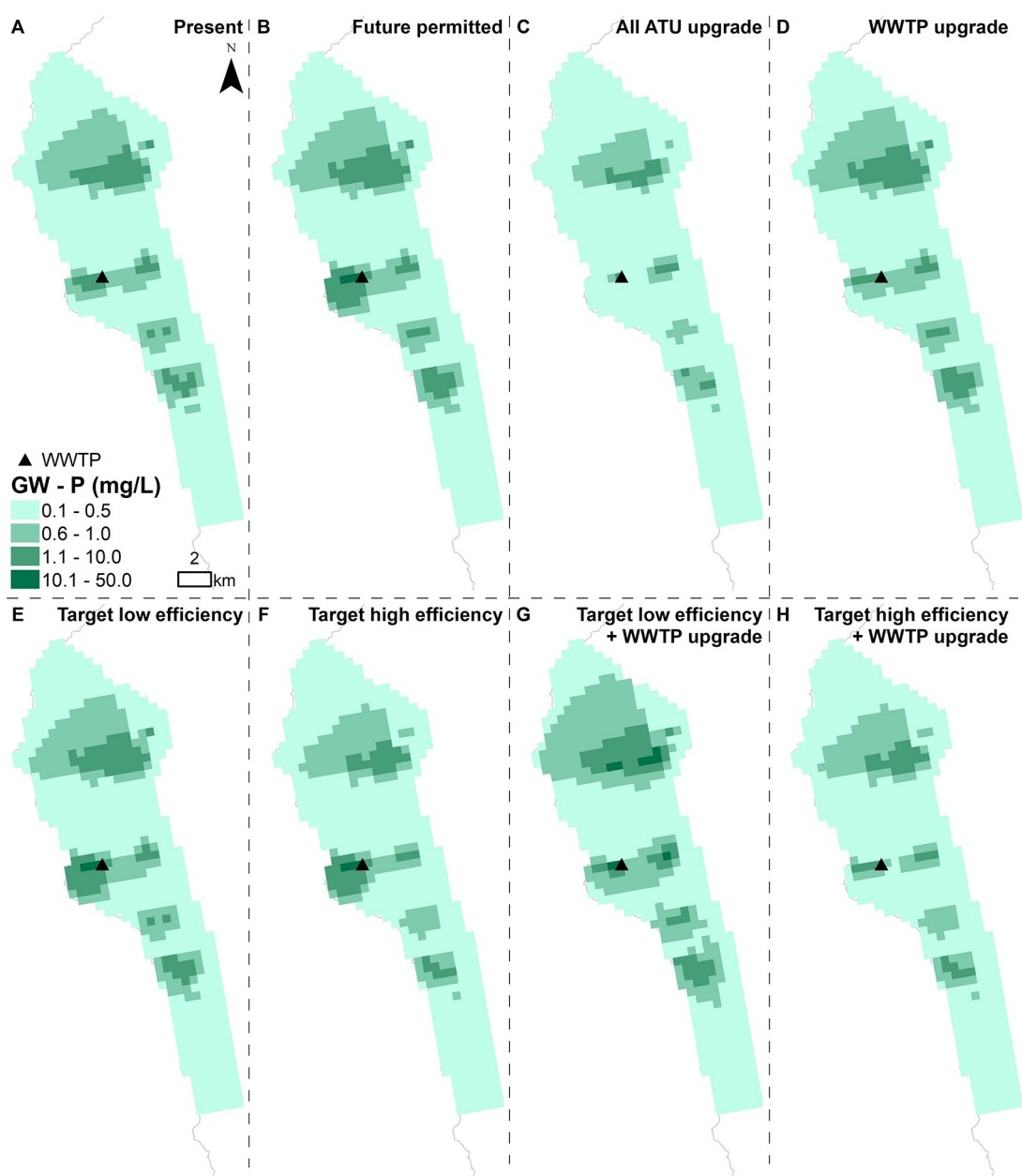

**Fig 4. Simulated groundwater P concentrations for (A) current conditions and (B-H) all scenarios.** Maps display results from the first layer of the groundwater model, where the bottom of the layer is set to 1 m below mean sea level.

marine habitat in the immediate vicinity and to the south of the WWTP is more exposed to nutrient load reduction. Upgrading the WWTP also benefits the marine habitat located downstream from it (Fig 7D and 7G–7H). The habitat to the north of the site is less exposed to change in nutrient load discharge due to less SGD.

### 3.3 Economic costs

The present value or life cycle cost of cesspool conversion for each scenario was added to the estimated WWTP upgrade cost (where applicable) to determine the total present value cost of each management option (Fig 8).

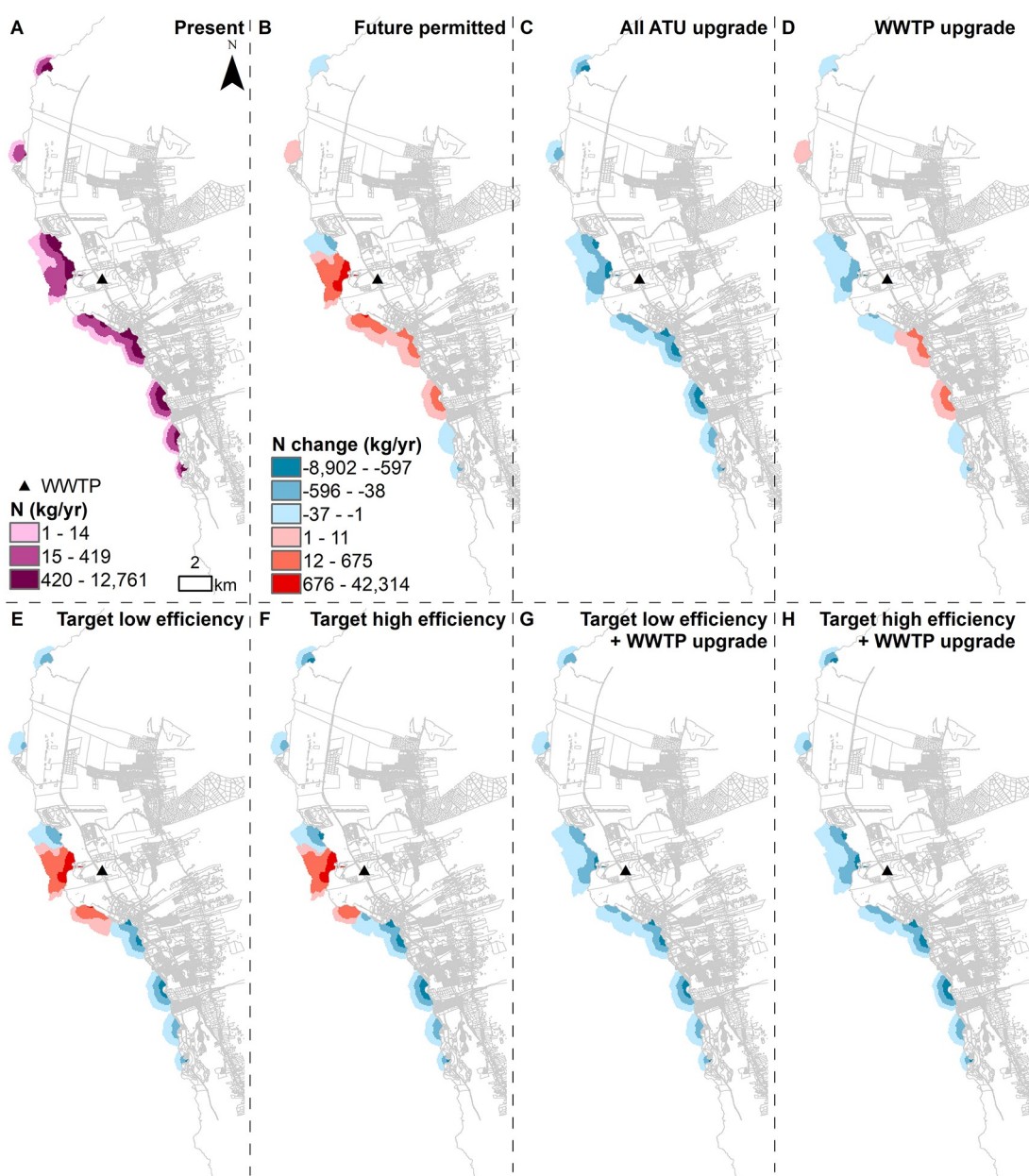

**Fig 5. Marine water quality.** Maps display (A) absolute loads of nitrogen (kg/year) under present conditions and (B-H) change (kg/year) relative to present conditions under all scenarios classified by geometrical intervals.

The present and future permitted build out scenarios incur zero costs because both assume that neither the cesspools nor the WWTP are upgraded. The WWTP upgrade scenario has the next lowest cost, requiring $160 million. The targeted cesspool conversion scenarios are roughly 67% more expensive with total costs of $267 million each, while the targeted cesspool conversion + WWTP upgrade scenarios cost $427 million, more than double the cost of only upgrading the WWTP. The all-ATU scenario is the most expensive, requiring $569 million, or approximately 3.5 times the cost of upgrading the WWTP alone.

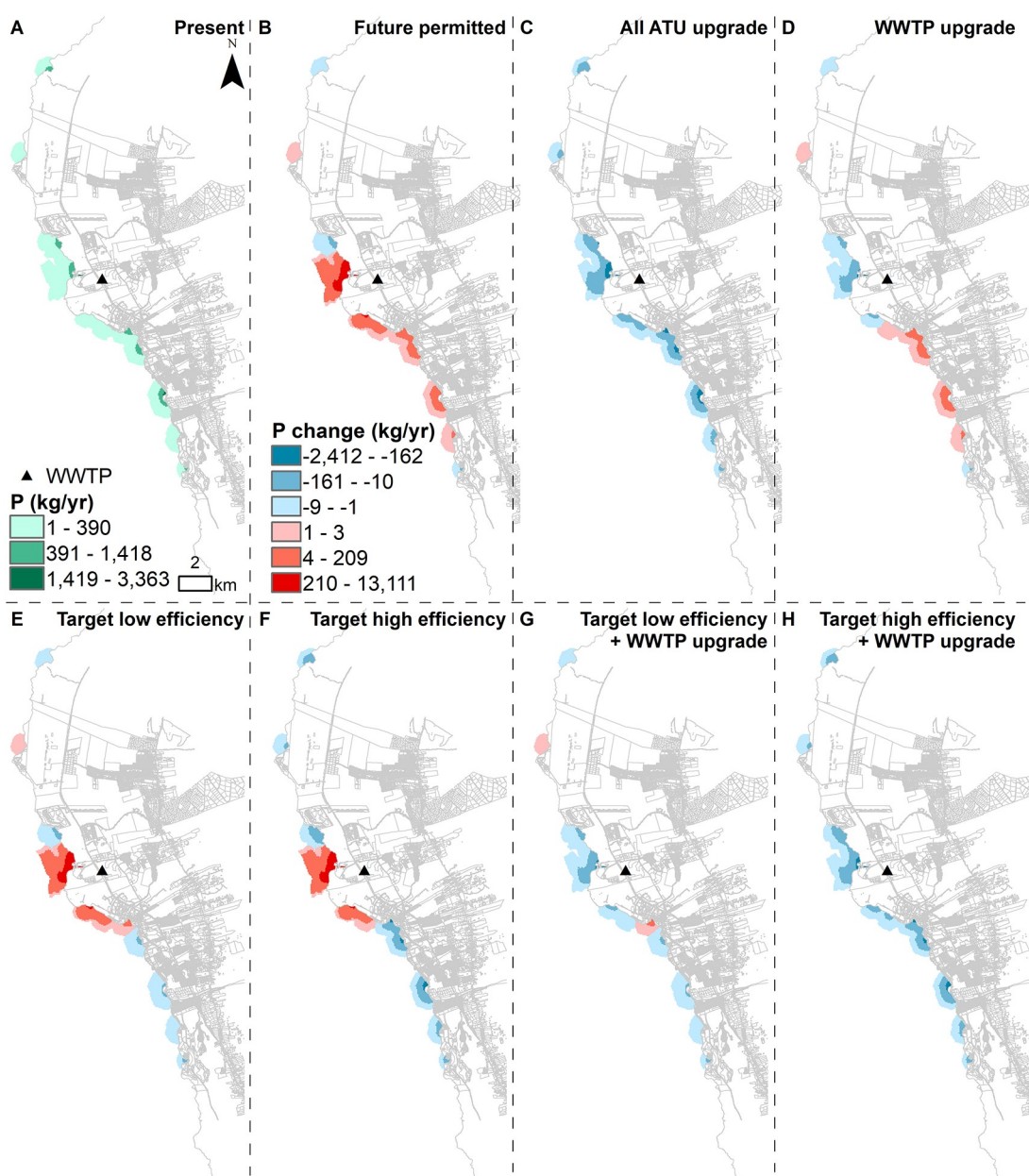

**Fig 6. Marine water quality.** Maps display (A) absolute loads of phosphorus (kg/year) under present conditions and (B-H) change (kg/year) relative to present conditions under all scenarios classified by geometrical intervals.

### 3.4 Tradeoffs

While it has been recognized that eliminating nutrient runoff significantly improves water quality and coastal ecosystem health, complete nutrient runoff elimination comes at a cost, often exceeding available funds. Tradeoff analysis can help inform what improvements can be achieved at different costs. Spatially aggregated changes in N, P, marine habitat quality, and present value cost relative to the current scenario are presented in Table 4. As expected, the largest increases in N (164 kg/d) and P (52 kg/d) occurred under the future permitted build out scenario, wherein future development added to the WWTP load and neither the cesspools

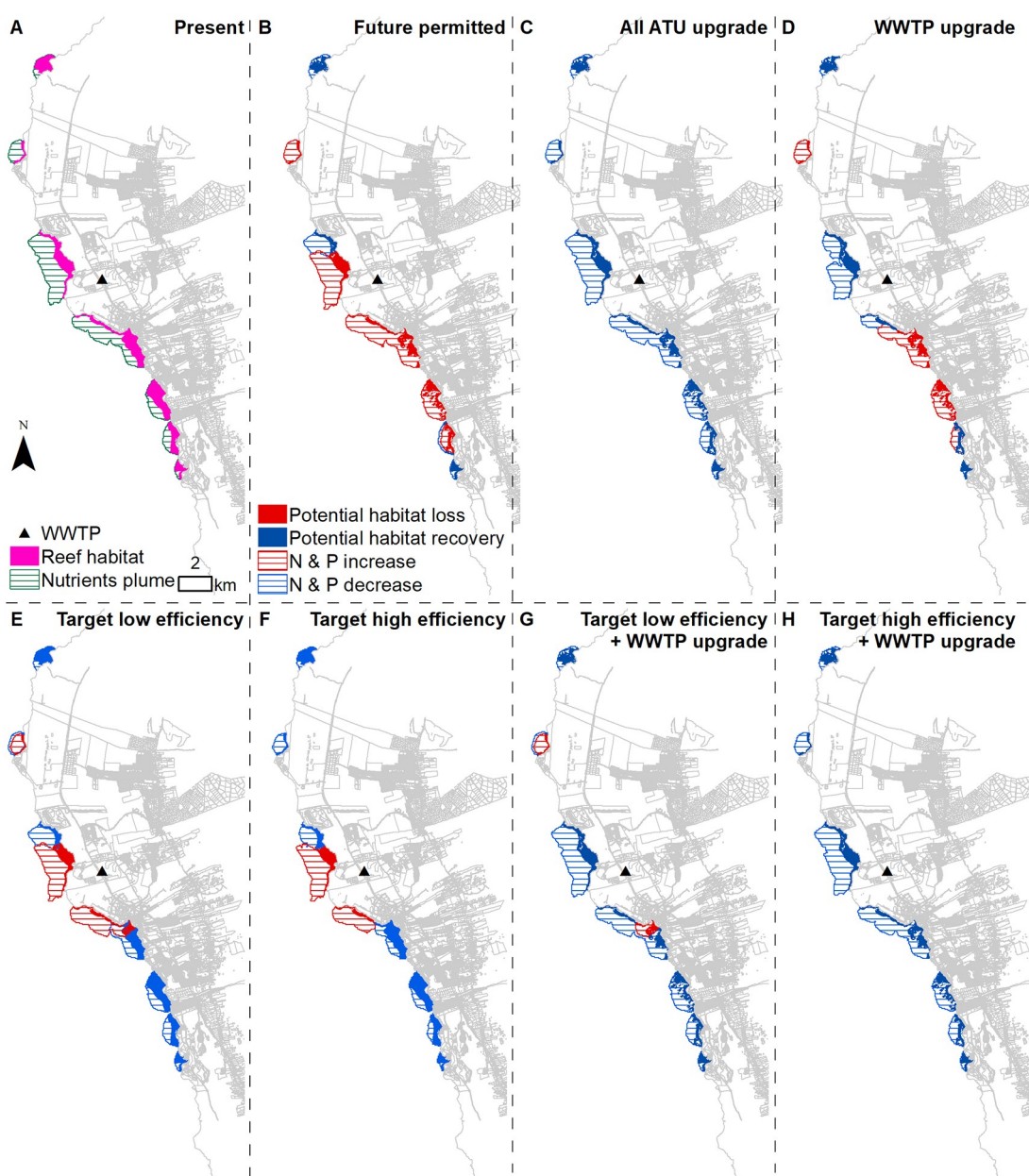

**Fig 7. Marine habitat potential impacts.** Maps display (A) the current exposure of marine habitat to SGD nutrients and (B-H) potential habitat loss or recovery due to the change in water quality under all scenarios.

nor the WWTP were upgraded. At the other end of the spectrum, N (-92 kg/d) and P (-15 kg/d) decreased the most when all cesspools were upgraded to ATUs (the most effective nutrient removal technology considered) and the WWTP was upgraded. Changes in habitat quality were similarly highest for the all-ATU scenario (666 ha) and lowest for the future permitted buildout scenario (-319 ha). The highest benefit outcomes, in terms of nutrient reduction and potential improvement in marine habitat quality, come at a cost, however. Upgrading all cesspools to ATUs has a price tag of $569 million, or at least double the cost of most other management options evaluated.

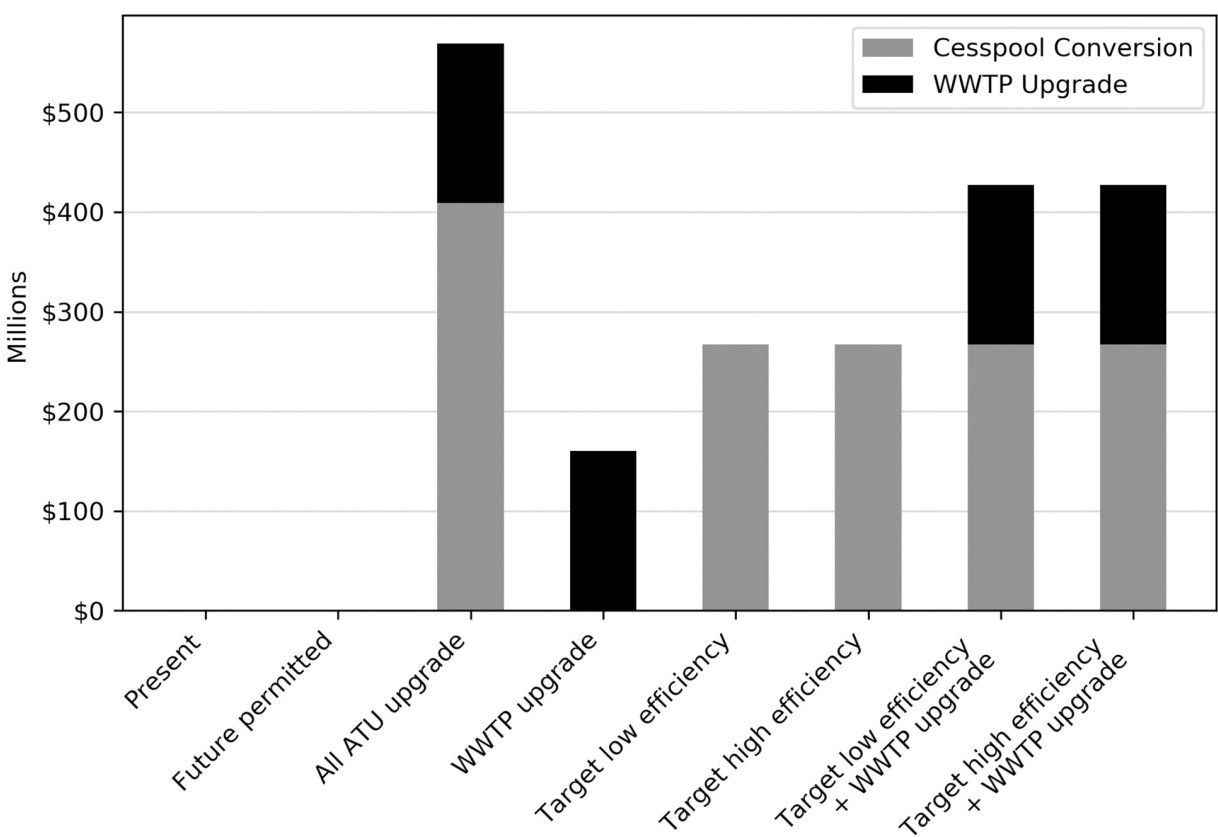

**Fig 8. Present value costs of cesspool and WWTP upgrades.**

To aid in the simultaneous interpretation of the four metrics across the seven potential future scenarios, spider diagrams were generated using the ordinal rankings of scenarios within each metric category. The future permitted build out scenario ranked highest for the cost metric but lowest for all of the environmental metrics, and therefore generated the smallest area (equivalently, overall ranking) among the seven scenarios evaluated (Fig 9G). At the other extreme, converting all cesspools to ATUs while simultaneously upgrading the WWTP was the most expensive but scored the highest for N reduction, P reduction, and marine habitat quality, resulting in the highest overall ranking (Fig 9A). The remaining five scenarios fell somewhere in between (Fig 9B–9F). Of note, the rankings were more sensitive to the WWTP upgrade than they were to the cesspool upgrade efficiency assumptions. In terms of cost-

**Table 4. Changes in N, P, marine habitat quality, and present value cost relative to the current scenario.**

| Scenario | ΔN | ΔP | ΔHabitat | PV Cost |
|---|---|---|---|---|
| | (kg/d) | (kg/d) | (ha) | (million $s) |
| Future permitted | 164 | 52 | -319 | 0 |
| All ATU upgrade | -92 | -15 | 666 | 569 |
| WWTP upgrade | -12 | -3 | -10 | 160 |
| Target low efficiency | 134 | 51 | 339 | 267 |
| Target high efficiency | 111 | 46 | 403 | 267 |
| Target low efficiency + WWTP upgrade | -42 | -4 | 474 | 427 |
| Target high efficiency + WWTP upgrade | -66 | -8 | 652 | 427 |

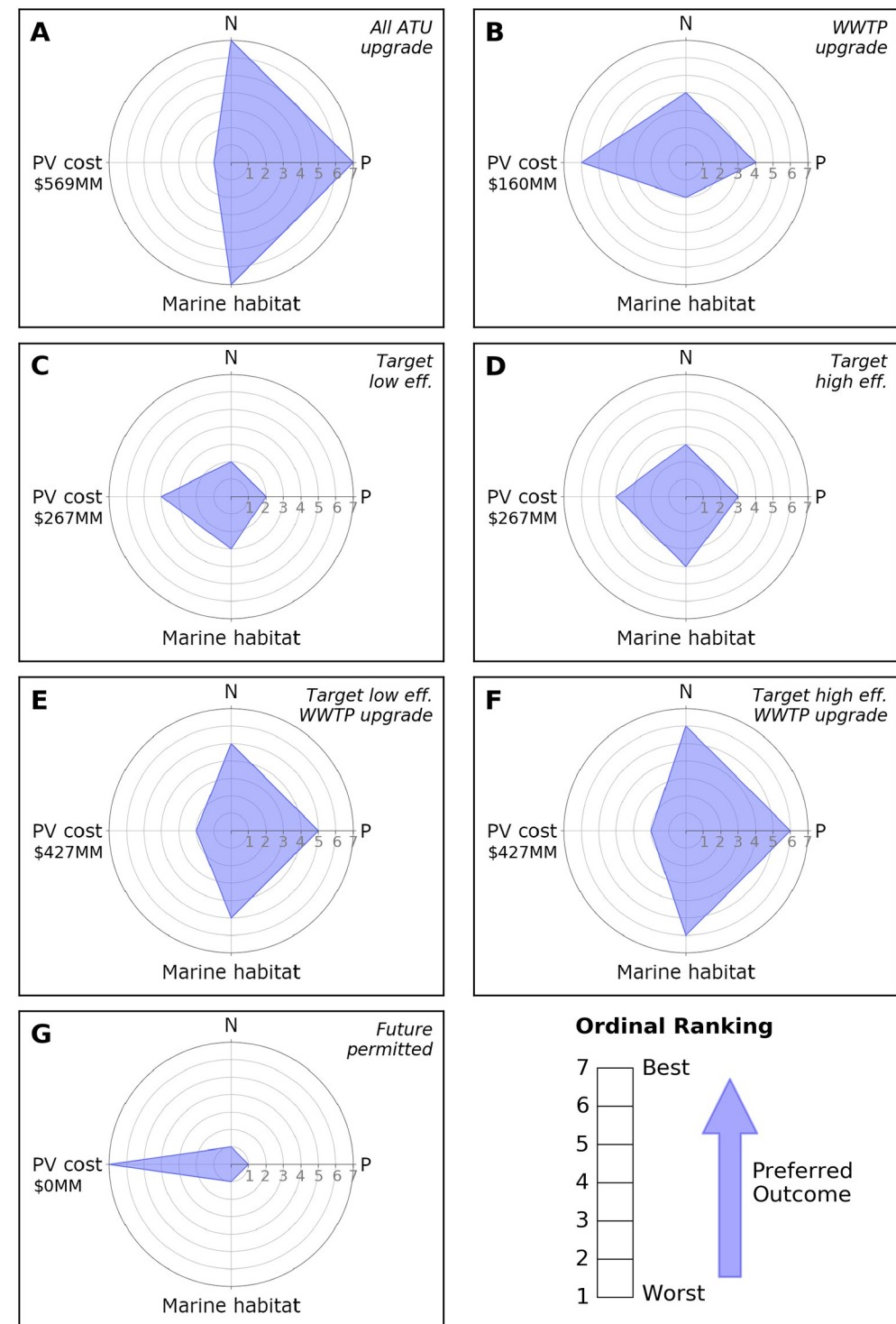

**Fig 9. Spider diagrams for each management scenario illustrating tradeoffs between N reduction, P reduction, potential change in marine habitat quality, and present value cost.**

**Table 5. Return on investment for each management strategy in terms of N, P, and habitat quality (change in physical units per million dollars invested).**

| Scenario | N (kg/d/$MM) | P (kg/d/$MM) | Habitat (ha/$MM) |
|---|---|---|---|
| All ATU upgrade | -0.451 | -0.117 | 1.732 |
| WWTP upgrade | -1.103 | -0.342 | 1.931 |
| Target low efficiency | -0.113 | -0.004 | 2.460 |
| Target high efficiency | -0.202 | -0.020 | 2.701 |
| Target low efficiency + WWTP upgrade | -0.483 | -0.130 | 1.854 |
| Target high efficiency + WWTP upgrade | -0.539 | -0.140 | 2.272 |

effectiveness, upgrading only the WWTP (Fig 9B) achieved better total nutrient reduction than targeted cesspool conversion (Fig 9C and 9D) at a fraction of the cost. However, that nutrient reduction did not correspond to a higher ranking in overall marine habitat quality because all of the nutrient reduction occurred only in the immediate vicinity of the WWTP, while nutrient reduction under targeted cesspool conversion was more evenly distributed along the coast.

The ROI for N was lowest for the target low efficiency scenario (0.113 kg/d reduction per million dollars) and highest for the WWTP upgrade scenario (1.103 kg/d reduction per million dollars). The targeted conversion scenarios with the WWTP upgrade (0.483–0.539 kg/d reduction per million dollars) generally tended to have higher ROI than their counterparts without the WWTP upgrade (0.113–0.202 kg/d reduction per million dollars) due to the relatively high benefit-cost ratio of the WWTP upgrade. Although the all-ATU upgrade scenario generated the largest simulated N-reduction benefit in absolute terms, its high cost resulted in a below-median ROI (0.451 kg/d reduction per million dollars). ROI rankings across scenarios were similar for P, ranging from a 0.004 kg/d reduction (target low efficiency) to a 0.342 kg/d reduction (WWTP upgrade) per million dollars. However, the ROI for potential habitat quality improvement followed a different pattern. Because the habitat quality benefits of the WWTP upgrade were concentrated within a relatively small area, the overall ROI for each scenario was less affected by whether the WWTP upgrade occurred or not. Thus, the target high efficiency scenario had the highest ROI for potential habitat quality improvement (2.701 ha per million dollars), while the all ATU upgrade scenario had the lowest (1.732 ha per million dollars). ROI values for all environmental metrics across each scenario are summarized in Table 5.

## 4. Discussion and conclusions

The choice of wastewater management scenario will involve tradeoffs among cost, nutrient loading, and marine outcomes. While a full upgrade and highest-level conversion plan will guard against the most nutrients and provide the highest protection of the near shore marine environment measured here, at $569 million in present value terms, this scenario is twice as expensive as following the targeted cesspool upgrade plan (without upgrading the WWTP) and over $100 million more expensive than the targeted cesspool conversion approach, including the WWTP upgrade. While these tradeoffs express the cumulative effect of the scenarios on these dimensions, spatial differences in impacts are obscured by these comparisons. For example, potential habitat loss from upgrading the WWTP but leaving the cesspools in place is geographically very different from the potential consequence of upgrading the cesspools but leaving the WWTP as is. Upgrading the WWTP but leaving the cesspools in place moves potential marine habitat loss from the area surrounding the plant towards the south, while upgrading the cesspools while preserving the current WWTP technology shifts the majority of the potential marine ecological losses to the area immediately offshore of the WWTP. One

takeaway from this analysis might be that concentrating exclusively on the cesspool conversions (per the language in Act 125) without concurrently improving the WWTP where all future permitted development will likely flow through may lead to suboptimal nutrient reductions and marine habitat outcomes.

Although the focus on the benefit side is largely on the environmental changes associated with each management action evaluated, additional co-benefits may be generated such as job creation and other economic impacts in the wastewater management industry. Providing an exhaustive list of co-benefits and quantitative estimates for each is beyond the scope of this study. However, impacts to Hawai'i's economy, defined as the direct, indirect, and induced economic activities generated by the local expenditures required to support each wastewater management action are likely to be in the millions of dollars and thousands of jobs. When type II multipliers generated from the State's input-output model [61] for the waste management and remediation services industry are applied to the costs of each scenario, estimated economic impacts range from $206–919 million in business sales, $55–247 million in earnings, $11–51 million in state tax revenue, and 980–4,371 jobs, depending on the management action and the share of total cost that is counted as local expenditures. Including such co-benefits help to provide a more holistic picture of the wastewater management problem, but it should be noted that while costlier management options tend to generate higher economic impacts, more is not necessarily better; if such co-benefits are more rigorously estimated to inform the evaluation of the management actions under consideration, the opportunity cost or alternative use of funds for each action should also be evaluated.

Not all possible combinations of wastewater treatment technologies are considered here, as current Hawai'i OSDS procedures, design criteria, standards, and restrictions are limited to septic tanks, ATUs, and absorption trenches/beds. More cost-effective and efficient alternatives have been implemented in other areas of the country and beyond, but such systems must be approved on a case-by-case basis by the State Department of Health. For example, layered soil treatment systems are a low maintenance disposal option with expected nitrogen removal in the range of 50 to 90 percent [24]. If paired with a standard septic tank, nutrient removal efficiency could be on par with an ATU system at a fraction of the cost in certain areas. In some instances, decentralized cluster wastewater systems may also be a more cost-effective alternative than individual OSDS upgrades. However, there are several factors to consider: the number of (current) cesspool owners in the cluster that would connect to the decentralized wastewater system, terrain, land availability, and the funding mechanism [24]. Because the optimal solution would likely require a portfolio of treatment technologies, one might interpret the results of this analysis as a base case that can be expanded upon in the future to consider emerging technologies as knowledge of such systems improve and as the state's OSDS technology approval procedures continue to evolve.

When a suboptimal choice is selected among the kinds of scenarios presented here, there can be tangible costs associated with reductions in coastal water and habitat quality, such as lower occupancy rates for areas like our study site in Hawai'i that are highly dependent on tourism. Perhaps more significant for local residents are the intangible costs such as unpleasant odors, unappealing windrows of invasive species biomass that are costly nuisances and biological threats to native marine biota. Such costs can be on the order of tens of millions of dollars or more. For example, results from a study of the Kīhei coast on Maui (Hawai'i) suggest that a failure to undertake nutrient abatement measures at the study site will eventually reduce current annual benefits derived from coral reefs by nearly $15 million [62]. The study further suggests that reducing nutrient levels will not only avoid losses but will also eventually increase annual benefits above the current level to nearly $30 million more than what would be realized in the absence of nutrient management. However, because of the delay between the time of

potential management interventions and reduction of nutrient levels in Kīhei, annual benefits will likely decline for at least a decade while the reef takes time to recover [62]. Long-term eutrophication of coasts that may result from the selection of cheaper but less effective wastewater management strategies sets up situations for likely invasive macroalgal and/or turf algal blooms [63, 64], loss of coral health [65], and increases in pathogenic microbial communities [66], changing profoundly the reef communities that draw people to live in coastal regions.

Although we did not explicitly model the potential impact of change in nutrient exposure on the nearshore habitat, wastewater impact from cesspools on coral reefs has been recognized as a major driver of groundwater and nearshore water quality degradation [5, 67, 68]. An increase in land-based nutrient exposure can promote benthic algae growth and hinder corals [48, 52, 53]. For example, a study in the Pacific region has shown that turf algae have an advantage over corals by becoming a fast colonizer in nutrient enriched conditions [55]. Adverse wastewater effect on coral reefs have also been identified around the tropics, including the Caribbean [69], the Red Sea [70], the Indian Ocean [71], the Florida Keys [72], and the Great Barrier Reef [73]. Therefore, wastewater management can reduce nutrient exposure of coral reefs and thereby minimize turf and macroalgae growth, providing space for coral recruits. This can also promote coral recovery post-bleaching events, especially in dry regions or shallow back-reef areas with limited water circulation [74], like the Kona coast. Additionally, it is important to note that whereas this study used mean annual values of nitrogen inputs to the coastline, future research could usefully explore temporal tidal variation, which is likely to have an important influence on algal growth rates.

While this work does not explicitly predict macroalgal blooms from elevated nutrients in SGD alone, it is important to consider how the combined impacts of nutrients and temperature increases due to wastewater management, climate change (e.g., increased sea surface temperatures), and fishing pressure may lead to decreases in herbivore abundance and accelerated growth of invasive macroalgae. A potential shift towards turf-algal dominated systems is likely in the scenarios where nutrient inputs are increased. The combination of decreased herbivore pressure and increased nutrients could lead to widespread, and ecologically disruptive macroalgal blooms, as seen on nearby Maui and Oʻahu, with severe negative impacts to charismatic-herbivores such as the green sea turtle [5, 52–54, 63, 65, 75]. This point is of special concern in areas with frequent inter-island and international boat traffic, such as that seen near the WWTP at Honōkohau Harbor, as boat traffic provides likely vectors for invasive algal introduction. Cascading effects of declining nearshore water quality may also include decreased water clarity from increased phytoplankton abundance, decreases in coral resilience to bleaching, as well as decreased fish and coral quantity and diversity [4, 19]. These ecological changes are linked to reduced human enjoyment of some of Kona's most prominent activities for residents and tourism industries, including nearshore snorkeling, scuba diving, shore fishing and spearfishing. Because the long-term biological consequences of excessive nutrient levels are generally not immediately reversible, careful consideration of such impacts is critical for coastal planning and management.

## Supporting information

**S1 Fig. Histogram of OSDS risk scores across the Keauhou basal aquifer.** Risk scores were calculated by [25].
(DOCX)

**S2 Fig. Numerical model grid setup for the Keauhou basal aquifer.** The vertical cross section A-A' consists of 21 layers. The top layer follows the local topography and bathymetry and the bottom layer follows a flat elevation of 550 m below mean sea level. The bottom elevation of

the top layer is set to 1 m below mean sea level to ensure dry cells were not produced. The layer thickness gradually increases, where the uppermost layers are thinnest.
(DOCX)

**S1 Table. Number of cesspools converted by type of upgrade for each management scenario.**
(DOCX)

## Acknowledgments

We are grateful to Stuart Coleman and Michael Mezzacapo of the Cesspool Conversion Working Group for continuing discussions about the state's plans for achieving the goals set out in Act 125. We gratefully acknowledge Christin Reynolds, Roger Babcock, Rick Bennet, and partners at Wastewater Alternatives and Innovations (WAI) for sharing their knowledge about wastewater management policy, technologies, and costs. We also thank Kā'eo Duarte for providing helpful feedback throughout the project. The views expressed are those of the author(s) and do not necessarily reflect the views of any of the agencies listed.

## Author Contributions

**Conceptualization:** Christopher A. Wada, Kimberly M. Burnett, Brytne K. Okuhata, Jade M. S. Delevaux, Leah L. Bremer.

**Data curation:** Christopher A. Wada, Brytne K. Okuhata, Jade M. S. Delevaux.

**Formal analysis:** Christopher A. Wada, Brytne K. Okuhata, Jade M. S. Delevaux.

**Funding acquisition:** Leah L. Bremer.

**Investigation:** Christopher A. Wada, Brytne K. Okuhata, Jade M. S. Delevaux.

**Methodology:** Christopher A. Wada, Brytne K. Okuhata, Jade M. S. Delevaux, Henrietta Dulai, Aly I. El-Kadi, Leah L. Bremer.

**Project administration:** Christopher A. Wada, Kimberly M. Burnett, Leah L. Bremer.

**Resources:** Henrietta Dulai, Aly I. El-Kadi, Leah L. Bremer.

**Software:** Brytne K. Okuhata, Jade M. S. Delevaux, Aly I. El-Kadi.

**Supervision:** Kimberly M. Burnett, Leah L. Bremer.

**Validation:** Brytne K. Okuhata, Jade M. S. Delevaux, Henrietta Dulai, Aly I. El-Kadi.

**Visualization:** Christopher A. Wada, Brytne K. Okuhata, Jade M. S. Delevaux.

**Writing – original draft:** Christopher A. Wada, Kimberly M. Burnett, Brytne K. Okuhata, Jade M. S. Delevaux.

**Writing – review & editing:** Christopher A. Wada, Kimberly M. Burnett, Brytne K. Okuhata, Jade M. S. Delevaux, Henrietta Dulai, Aly I. El-Kadi, Veronica Gibson, Celia Smith, Leah L. Bremer.

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
