## [Decision Letter · Decision Letter 0]

26 Jul 2021

PONE-D-21-17746

Identifying wastewater management tradeoffs: costs, nearshore water quality, and implications for marine coastal ecosystems in Kona, Hawai‘i

PLOS ONE

Dear Dr. Burnett,

Thank you for submitting your manuscript to PLOS ONE. After careful consideration, we feel that it has merit but does not fully meet PLOS ONE’s publication criteria as it currently stands. Therefore, we invite you to submit a revised version of the manuscript that addresses the points raised during the review process.

ACADEMIC EDITOR:

Thank you for submitting an interesting and unique study, spanning from groundwater to coral reefs. I agree with the reviewers that more work is needed on the manuscript, and decide on a major revision.

We look forward to receiving your revised manuscript.

Kind regards,

Andreas C. Bryhn

Academic Editor

PLOS ONE

Journal Requirements:

Funding for this project came from the National Science Foundation’s Research Infrastructure Improvement Award (RII) Track-1: ‘Ike Wai: Securing Hawai‘i’s Water Future Award #OIA-1557349 and USGS Water Resources Research Institute Program grant number G16AP00049 BY5 “Linking watershed and groundwater management to groundwater dependent ecosystems and their linked ecological, cultural, and socio-economic values.”

3. We note that Figures 1 and 2 in your submission contain [map/satellite] images which may be copyrighted. All PLOS content is published under the Creative Commons Attribution License (CC BY 4.0), which means that the manuscript, images, and Supporting Information files will be freely available online, and any third party is permitted to access, download, copy, distribute, and use these materials in any way, even commercially, with proper attribution. For these reasons, we cannot publish previously copyrighted maps or satellite images created using proprietary data, such as Google software (Google Maps, Street View, and Earth). For more information, see our copyright guidelines: http://journals.plos.org/plosone/s/licenses-and-copyright.

a. You may seek permission from the original copyright holder of Figures 1 and 2 to publish the content specifically under the CC BY 4.0 license.  

Reviewers' comments:

Reviewer's Responses to Questions

**Comments to the Author**

1. Is the manuscript technically sound, and do the data support the conclusions?

Reviewer #1: Yes

Reviewer #2: Yes

2. Has the statistical analysis been performed appropriately and rigorously? 

Reviewer #1: Yes

Reviewer #2: Yes

3. Have the authors made all data underlying the findings in their manuscript fully available?

Reviewer #1: Yes

Reviewer #2: Yes

4. Is the manuscript presented in an intelligible fashion and written in standard English?

Reviewer #1: Yes

Reviewer #2: Yes

5. Review Comments to the Author

Reviewer #1: In this manuscript, the authors aim to assess wastewater management scenarios and trade-offs for improving water quality on coral reefs in Kona, Hawaii. The question is relevant and interesting, and the dataset and modelling approach are impressive. In general, I find the text to be well-writte, albeit could improve in conciseness/clarity, and commend the authors for the work that obviously went into this manuscript. My biggest concern involves the trade-off analysis, along with some of the terminology to be misleading, and some more minor clarification needed. Please find my major and minor comments below.

Major comments:

Most of my major concerns involve the trade-off analysis. I find the spider diagram/ranking approach to be a bit convoluted. It seems that all data is available to do (e.g., Table 4) a proper cost-benefit analysis and it is unclear to me why this was not done? While the spider diagrams are potentially useful as a supplemental figure, determining the benefit (e.g., improvement in water quality) per unit cost of each wastewater management scenario seems a more straightforward and justifiable approach. This would provide a clearer, more easily explainable prioritisation of which scenario provides the greatest benefit per dollar, allowing for more concrete recommendations. Beher et al. 2016 Prioritising catchment management projects to improve marine water quality exemplifies the sort of approach I am referring to in a similar, water quality context.

I find the framing of “coral reef habitat impact” to be misleading – impact, to me, implies a change in condition. As the authors do not measure or model any change in coral reef condition due to nutrient exposure (which is difficult to do!) I would suggest removing reference to impact and framing around nutrient exposure or potential risk of impact on coral reefs. Some discussion around the fact that we really don’t know much about how corals would respond would be useful in the limitations section of the discussion.

Minor comments:

General minor comments:

1. Ensure that acronyms are all defined when they are presented

2. I think in many cases the writing could be a bit more concise – a read for conciseness

3. Check citation formatting as some intext citations were not clear – but best to check journal guidelines (maybe it’s correct?).

Line 58: the phrasing “most important” is subjective – I would consider rewording

Line 93: Can you give an example of a ground-water dependent ecosystem? I am not as familiar so this would help clarify what the authors are talking about

Line 133: Very minor, but wondering why it is referred to as “mountain-to-sea” as opposed to the more common “ridge-to-reef” framing.

Line 154: Capitalise “Scenarios”

Line 164: I find the “risk scores” aspect described in the methods to be very unclear currently. How were they determined? Is the 12 threshold (line 215) basically arbitrary? What are the implications of some of the decisions made around the risk scores and how might they change results?

Table 1. I wish there was more detail about the scenarios here so you could understand them from just the table quickly instead of trying to dig through the methods. For example – for the All ATU upgrade, I thought it would be All-ATU under cesspool conversion and a N under WWTP upgrade – or is the “upgrade”referring to WWTP? Maybe the name is misleading? The difference between Scenario 1 and 2 is also confusing until you read through the text. In general, the table brought up a lot of confusion for me until I read through the text, but even then some aspects weren’t particularly clear. A quick description column would help.

Line 248: I assume the 21 layers are spatial layers? Is there a supplemental table with a description of these?

Line 286: Why was 1.2 km from the shoreline chosen in the diffusion model? A brief justification would help.

Line 327: was confused what BR stood for – it is introduced later – make sure introduced where first mentioned.

Line 502: Beyond the scope of this paper – but just a note that it seems this analysis could be upgraded fairly easily to a spatial prioritisation. This would be very interesting extension of the work that I hope the authors look into!

Line 549: “A shift towards turf-algal dominated systems is seen in the scenarios where nutrient inputs are increased” – this seems like it came out of no where? I don’t see any mention of measuring changes in benthic cover from nutrients, from what I can tell it is only changes in nutrient concentrations so then I am confused on how shifts to turf-dominated systems are detected?

Figures – In general the figures are very nicely done and laid out. For the figures describing nutrient decreases/increases – why are they binned into high/medium/low? What values are considered in each of these bins/how were they determined? Why not sure a continuous scale or quartiles with actual values?

Reviewer #2: While the modeling exercise in this manuscript sets out to answer the question of which range of wastewater treatment has met that goal I feel that it could be improved by also setting the stage for what the value of a healthier reef system would be an to also run a no action scenario which show the costs incurred or income lost by inaction.

To be specific. In the introduction the authors could more specifically call out income (tourism, home values, etc) generated by healthy reefs as well as long term coastal protection by a reef more resilient in the face of sea level rise and climate change. There should also be a clear connection to the costs of high levels of nitrates in drinking water wells which under no action will be costly to remove and is costly due to health risks such as colorectal cancers, blue baby syndrome etc. The NIH has good reports out on this recently.

This information could then be reinforced in the cost benefit discussion in the conclusion. Where the reader would then be able to clearly see a multi-million dollar investment could save X millions of dollars in revenue, and risk avoidance annually.

Additional co-benefits should also be researched such as job creation as decentralized wastewater has the opportunity to create hundreds if not thousands of jobs across multiple sectors (design/engineering, technical installers, service providers, supply chain and distribution, and finally regulatory).

I do believe the authors achieved what they set out to do but the above amendments which are mostly well documented could help to reinforce the purpose for the larger investment recommended.

In this layout it would almost read like a Environmental Impact Statement where a No action alternative should also be played out along with the other alternatives.

6. PLOS authors have the option to publish the peer review history of their article (what does this mean?). If published, this will include your full peer review and any attached files.

Reviewer #1: No

Reviewer #2: **Yes: **Christopher S Clapp

---

## [Author Response · Author response to Decision Letter 0]

23 Aug 2021

The style guide has been followed, thank you.

Funding for this project came from the National Science Foundation’s Research Infrastructure Improvement Award (RII) Track-1: ‘Ike Wai: Securing Hawai‘i’s Water Future Award #OIA-1557349 and USGS Water Resources Research Institute Program grant number G16AP00049 BY5 “Linking watershed and groundwater management to groundwater dependent ecosystems and their linked ecological, cultural, and socio-economic values.”

Now amended in cover letter, thank you.

3. We note that Figures 1 and 2 in your submission contain [map/satellite] images which may be copyrighted. All PLOS content is published under the Creative Commons Attribution License (CC BY 4.0), which means that the manuscript, images, and Supporting Information files will be freely available online, and any third party is permitted to access, download, copy, distribute, and use these materials in any way, even commercially, with proper attribution. For these reasons, we cannot publish previously copyrighted maps or satellite images created using proprietary data, such as Google software (Google Maps, Street View, and Earth). For more information, see our copyright guidelines: http://journals.plos.org/plosone/s/licenses-and-copyright.

Thank you for pointing those out. For Fig 1: 

- inset A and B shows maps obtained from Fukunaga & Associates [37] and publicly available; 

- inset C shows aggregated OSDS generated by Okuhata et al. [32], and is publicly available, while the location of the WWTP is also publicly available since 2019 (see: https://geoportal.hawaii.gov/datasets/662795ebbc20438aa66aa150df7b1e2a/explore?location=20.661397%2C-157.362250%2C7.85 ); 

- inset D shows data generated by [37] and for this project; and 

- inset E shows the publicly available marine habitat map (https://coastalscience.noaa.gov/project/benthic-habitat-mapping-main-hawaiian-islands/), which we cite. 

For Fig 2, all the maps shown are from either already published sources which cite in the body of the manuscript or data generated for the project, except for the wave model image. 

We obtained the copyright release from the owner and provided the permission in the “Other” section of the PLOS ONE upload system.

Done!

Thank you, we have now included the caption for our two new supporting figures (Fig SI and S2) and one table (Table S1) in our Supporting Information at the end of our manuscript.

Comments to the Author

1. Is the manuscript technically sound, and do the data support the conclusions?

Reviewer #1: Yes

Reviewer #2: Yes

2. Has the statistical analysis been performed appropriately and rigorously? 

Reviewer #1: Yes

Reviewer #2: Yes

3. Have the authors made all data underlying the findings in their manuscript fully available?

Reviewer #1: Yes

Reviewer #2: Yes

4. Is the manuscript presented in an intelligible fashion and written in standard English?

Reviewer #1: Yes

Reviewer #2: Yes

5. Review Comments to the Author

Reviewer #1: In this manuscript, the authors aim to assess wastewater management scenarios and trade-offs for improving water quality on coral reefs in Kona, Hawaii. The question is relevant and interesting, and the dataset and modelling approach are impressive. In general, I find the text to be well-writte, albeit could improve in conciseness/clarity, and commend the authors for the work that obviously went into this manuscript. My biggest concern involves the trade-off analysis, along with some of the terminology to be misleading, and some more minor clarification needed. Please find my major and minor comments below.

Major comments:

Most of my major concerns involve the trade-off analysis. I find the spider diagram/ranking approach to be a bit convoluted. It seems that all data is available to do (e.g., Table 4) a proper cost-benefit analysis and it is unclear to me why this was not done? While the spider diagrams are potentially useful as a supplemental figure, determining the benefit (e.g., improvement in water quality) per unit cost of each wastewater management scenario seems a more straightforward and justifiable approach. This would provide a clearer, more easily explainable prioritisation of which scenario provides the greatest benefit per dollar, allowing for more concrete recommendations. Beher et al. 2016 Prioritising catchment management projects to improve marine water quality exemplifies the sort of approach I am referring to in a similar, water quality context.

Thank you for this suggestion. We agree that estimating benefits per dollar spent on management actions would allow for more concrete policy recommendations. The manuscript now includes ROI calculations, where benefits are measured in physical units and costs are measured in dollars. We also expanded the methods section to explain how the cost-benefit calculations were done.

I find the framing of “coral reef habitat impact” to be misleading – impact, to me, implies a change in condition. As the authors do not measure or model any change in coral reef condition due to nutrient exposure (which is difficult to do!) @I would suggest removing reference to impact and framing around nutrient exposure or potential risk of impact on coral reefs. Some discussion around the fact that we really don’t know much about how corals would respond would be useful in the limitations section of the discussion.

Thank you for highlighting this point. We agree and reframed the approach around potential risk of habitat impact throughout the manuscript. 

We also added a section in the discussion regarding the limitation of not modeling the response of marine habitat to change in nutrient exposure:

“Although we did not explicitly model the potential impact of change in nutrient exposure on the nearshore habitat, wastewater impact from cesspools on coral reefs has been recognized as a major driver of groundwater and nearshore water quality degradation [5, 68, 69]. An increase in land-based nutrient exposure can promote benthic algae growth and hinder corals [48, 53, 54]. For example, a study in the Pacific region has shown that turf algae have an advantage over corals by becoming a fast colonizer in nutrient enriched conditions [56]. Adverse wastewater effect on coral reefs have also been identified around the tropics, including the Caribbean [70], the Red Sea [71], the Indian Ocean [72], the Florida Keys [73], and the Great Barrier Reef [74]. Therefore, wastewater management can reduce nutrient exposure of coral reefs and thereby minimize turf and macroalgae growth, providing space for coral recruits. This can also promote coral recovery post-bleaching events, especially in dry regions or shallow back-reef areas with limited water circulation [75], like the Kona coast. Additionally, it is important to note that whereas this study used mean annual values of nitrogen inputs to the coastline, future research could usefully explore temporal tidal variation, which is likely to have an important influence on algal growth rates. ”

Minor comments:

General minor comments:

1. Ensure that acronyms are all defined when they are presented

2. I think in many cases the writing could be a bit more concise – a read for conciseness

3. Check citation formatting as some intext citations were not clear – but best to check journal guidelines (maybe it’s correct?).

Line 58: the phrasing “most important” is subjective – I would consider rewording

Agreed -- rephrased to “a significant” threat

Line 93: Can you give an example of a ground-water dependent ecosystem? I am not as familiar so this would help clarify what the authors are talking about

Right, these are common in Hawaii but not everywhere, we now provide wetlands and spring-fed agricultural systems as examples in the text.

Line 133: Very minor, but wondering why it is referred to as “mountain-to-sea” as opposed to the more common “ridge-to-reef” framing.

We agree here as well, while several of this paper’s co-authors have used this framing in past projects, the “ridge-to-reef” framing is much more common and probably more appropriate for this publication outlet.

Line 154: Capitalise “Scenarios”

Now capitalized.

Line 164: I find the “risk scores” aspect described in the methods to be very unclear currently. How were they determined? Is the 12 threshold (line 215) basically arbitrary? What are the implications of some of the decisions made around the risk scores and how might they change results?

Original risk scores were determined by Whittier and El-Kadi (2014). Using a geographical mapping and numerical modeling results, a weight and rating method was utilized to compute an overall risk score. Various input factors, such as proximity to shoreline, distance to groundwater, and soil type, were weighted based on their contribution to OSDS risk. The median risk score of 12 was selected as the conversion threshold, and we now provide a histogram in the Supporting Information Fig. S1. We recognize that the selected threshold can impact results due to the significant differences between the risk score frequencies. The number of existing OSDS units converted to ATU systems could range from 8% to 84%, depending on whether a risk score of 11, 12, or 13 was selected as the conversion threshold. This would ultimately change the total nutrient mass load that enters the aquifer and travels to the coastline. By selecting the median risk score as the conversion threshold, 60% of the existing OSDS units will be converted to ATU systems, which we believe is a realistic amount, thus producing practical results. 

Fig. S1. Histogram of OSDS risk scores across the Keauhou basal aquifer. Risk scores were calculated by Whittier and El-Kadi (2014).

Table 1. I wish there was more detail about the scenarios here so you could understand them from just the table quickly instead of trying to dig through the methods. For example – for the All ATU upgrade, I thought it would be All-ATU under cesspool conversion and a N under WWTP upgrade – or is the “upgrade”referring to WWTP? Maybe the name is misleading? The difference between Scenario 1 and 2 is also confusing until you read through the text. In general, the table brought up a lot of confusion for me until I read through the text, but even then some aspects weren’t particularly clear. A quick description column would help.

Thank you for pointing this out, we agree the scenarios as described in Table 1 are confusing. Following your suggestion, we have now updated the table with an additional column and have expanded and clarified the descriptions regarding the cesspool conversions and the WWTP upgrades for each scenario.

Line 248: I assume the 21 layers are spatial layers? Is there a supplemental table with a description of these?

The 21 layers are horizontal layers that follow the local topography and bathymetry and extend down to an elevation of 550 m below mean sea level. More information regarding the numerical model grid has been provided. Additionally, a figure displaying the model grid has been added to the supporting information.

Fig. S2 Numerical model grid setup for the Keauhou basal aquifer. The vertical cross section A-A’ consists of 21 layers. The top layer follows the local topography and bathymetry and the bottom layer follows a flat elevation of 550 m below mean sea level. The bottom elevation of the top layer is set to 1 m below mean sea level to ensure dry cells were not produced. The layer thickness gradually increases, where the uppermost layers are thinnest.

Line 286: Why was 1.2 km from the shoreline chosen in the diffusion model? A brief justification would help.

The spread of nutrient loads into the marine environment from each pourpoint was modeled using a decay function (see Equation 1), which assigned a portion of the remaining nutrient loads from the previous cell to all adjacent cells based on the diffusion factor layer until a maximum distance of 1.2 km (distance based on thermal infrared imagery from Johnson et al 2008) from the shoreline was reached (Halpern et al. 2008, Knee et al. 2010). This threshold was based on measurement of infrared imagery from the study site in ArcGIS and consultation with local experts.

Line 327: was confused what BR stood for – it is introduced later – make sure introduced where first mentioned.

The acronym BR (bedrooms) is now defined the first time it is introduced in the manuscript. 

Line 502: Beyond the scope of this paper – but just a note that it seems this analysis could be upgraded fairly easily to a spatial prioritisation. This would be very interesting extension of the work that I hope the authors look into!

We agree that a spatial prioritization would be the ideal type of analysis. It is somewhat possible to infer spatial preferences from the current analysis, for example due to high levels of SGD south of the wastewater treatment plant, cesspool conversion payoffs are larger across this broader area, while upgrades to the wastewater treatment plant alone enhance water quality only in the immediate vicinity of the plant. Our current analysis could be extended, perhaps with additional input and engagement from the surrounding community, to include a prioritization of wastewater treatment technologies based on priorities about which areas are most important to protect. Currently we can only provide ordinal rankings of technologies based on the specific metrics of interest, e.g., N or P reduction, present value cost, or marine habitat protection. 

Figures – In general the figures are very nicely done and laid out. For the figures describing nutrient decreases/increases – why are they binned into high/medium/low? What values are considered in each of these bins/how were they determined? Why not sure a continuous scale or quartiles with actual values?

Great point. We updated Fig 4 and 6 to display the actual values. We used the geometrical intervals from the best and worst case scenarios to identify the breaks and still highlight the differences across scenarios. We also added a sentence in the method section for the WQ model explaining this:

“The results displayed here use the geometrical intervals from the “best” (scenario 3, full upgrade) and “worst” (scenario 2; no upgrade) case scenarios to identify the breaks while also highlighting the differences across scenarios to enable comparison.”

Reviewer #2: While the modeling exercise in this manuscript sets out to answer the question of which range of wastewater treatment has met that goal I feel that it could be improved by also setting the stage for what the value of a healthier reef system would be an to also run a no action scenario which show the costs incurred or income lost by inaction.

We agree that the wastewater management analysis could have been better motivated by investigating the consequences to the reef system of a no action scenario. We do actually consider this scenario (scenario 2 -- the “future permitted” scenario) but did not adequately explain it as inaction, and did not adequately motivate the analysis by considering potential implications to the healthy reef system from inaction. We now clarify scenario 2 as the inaction scenario, and use this to better motivate the overall wastewater analysis. 

To be specific. In the introduction the authors could more specifically call out income (tourism, home values, etc) generated by healthy reefs as well as long term coastal protection by a reef more resilient in the face of sea level rise and climate change.

Thank you for this comment, we now specifically call out these values generated by healthy coral reefs in the introduction section of the paper: “Healthy coral reefs provide a suite of societal benefits to important income and livelihood sectors including by supporting fisheries and food security, and by enhancing recreation and tourism value [8, 9, 10]. Coral reefs also provide important coastal protection services, which enhances resilience to climate change [11, 12].”

We also added the following: “Such threats to coral reefs can have devastating impacts on coral reef ecology and the array of ecosystem services provided by these systems [8, 14]” and further elaborated: “Hawai‘i has become an important focal point for wastewater management within the United States due to increasing concern about threats to coral reef ecosystems of high value for tourism, recreation, fisheries, and cultural connection to place [15, 16]. Moreover, the connection between existing wastewater technologies and nutrient loading to the nearshore has also been extensively studied in Hawaiʻi [17, 18, 19, 20, 21, 22, 23].”

There should also be a clear connection to the costs of high levels of nitrates in drinking water wells which under no action will be costly to remove and is costly due to health risks such as colorectal cancers, blue baby syndrome etc. The NIH has good reports out on this recently. This information could then be reinforced in the cost benefit discussion in the conclusion. Where the reader would then be able to clearly see a multi-million dollar investment could save X millions of dollars in revenue, and risk avoidance annually.

We thank the reviewer for this suggestion. The highest observed nitrate levels in drinking water wells were up to ~1.5 mg/L, and up to 3 mg/L in coastal springs. While these concentrations are >100x above the ambient coastal ocean nitrate levels and therefore have a great potential to affect marine ecosystems, they are well below national drinking water standards. EPA considers concentrations below 5 mg/L, half of the maximum contaminant level (MCL) for nitrate set to protect against blue-baby syndrome, which is 10 mg/L, low health risk (https://www.epa.gov/nutrient-policy-data/estimated-nitrate-concentrations-groundwater-used-drinking). As such, a nitrate removal scenario targeting drinking water has not been deemed necessary.

Additional co-benefits should also be researched such as job creation as decentralized wastewater has the opportunity to create hundreds if not thousands of jobs across multiple sectors (design/engineering, technical installers, service providers, supply chain and distribution, and finally regulatory).

Thank you for this suggestion. We have now included a discussion of the local economic impacts of projected spending associated with potential management actions, in terms of job creation, business sales, earnings, and state tax revenue.

I do believe the authors achieved what they set out to do but the above amendments which are mostly well documented could help to reinforce the purpose for the larger investment recommended.

In this layout it would almost read like a Environmental Impact Statement where a No action alternative should also be played out along with the other alternatives.

6. PLOS authors have the option to publish the peer review history of their article (what does this mean?). If published, this will include your full peer review and any attached files.

Do you want your identity to be public for this peer review? For information about this choice, including consent withdrawal, please see our Privacy Policy.

Reviewer #1: No

Reviewer #2: Yes: Christopher S Clapp

---

## [Editor Report · Decision Letter 1]

25 Aug 2021

Identifying wastewater management tradeoffs: costs, nearshore water quality, and implications for marine coastal ecosystems in Kona, Hawai‘i

PONE-D-21-17746R1

Dear Dr. Burnett,

We’re pleased to inform you that your manuscript has been judged scientifically suitable for publication and will be formally accepted for publication once it meets all outstanding technical requirements.

Kind regards,

Andreas C. Bryhn

Academic Editor

PLOS ONE
---

## [Editor Report · Acceptance letter]

27 Aug 2021

PONE-D-21-17746R1 

Identifying wastewater management tradeoffs: costs, nearshore water quality, and implications for marine coastal ecosystems in Kona, Hawai‘i 

Dear Dr. Burnett:

I'm pleased to inform you that your manuscript has been deemed suitable for publication in PLOS ONE. Congratulations! Your manuscript is now with our production department. 

Kind regards, 

on behalf of

Dr. Andreas C. Bryhn 

Academic Editor

PLOS ONE